# MaxPoolBERT: Enhancing BERT Classification via Layer- and Token-Wise Aggregation

## Abstract

The `[CLS]` token in BERT is commonly used as a fixed-length representation for classification tasks, yet prior work has shown that both other tokens and intermediate layers encode valuable contextual information. In this work, we study lightweight extensions to BERT that refine the `[CLS]` representation by aggregating information across layers and tokens. Specifically, we explore three modifications: (i) max-pooling the `[CLS]` token across multiple layers, (ii) enabling the `[CLS]` token to attend over the entire final layer using an additional multi-head attention (MHA) layer, and (iii) combining max-pooling across the full sequence with MHA. Our approach, called MaxPoolBERT, enhances BERT's classification accuracy (especially on low-resource tasks) without requiring new pre-training or significantly increasing model size. Experiments on the GLUE benchmark show that MaxPoolBERT consistently achieves a better performance than the standard BERT base model on low resource tasks of the GLUE benchmark.

## 1 Introduction

BERT (Bidirectional Encoder Representations from Transformers) (Devlin et al., 2019), is one of the best known Transformer-based (Vaswani et al., 2017) language models. The core principle of BERT is the unsupervised pre-training approach on large corpora, enabling it to learn contextual word representations, which can then be used to solve various downstream tasks. Through fine-tuning, BERT adapts its representations to aggregate the most relevant information required for a given task.

A key component of BERT's architecture is the classification token (abbreviated `[CLS]`), a special token that is prepended to every input sequence. During fine-tuning, the `[CLS]` token serves as the only input to the classification head, which generates predictions for the task at hand. Through self-attention, the `[CLS]` token is expected to capture the sentence-level information necessary for downstream tasks. In this paper, we ask the question whether we can enrich the `[CLS]` token with information from the layers below the top level.

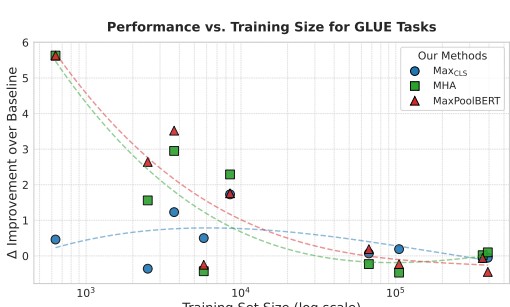

Figure 1: **MaxPoolBERT performs best on low-resource datasets.** We show that our methods, in particular MaxPoolBERT, provide significant improvements for smaller datasets indicating that the model learns a better representation during fine-tuning (top-left).

We know that the last layers of BERT change the most during fine-tuning and encode the most task-specific information (Rogers et al., 2020). This is why the `[CLS]` token embedding from the final layer is conventionally used for classification. However, assuming that only the `[CLS]` token retains meaningful sentence-level information is misleading. Prior studies have shown that all token embeddings in the final layer contain sentence-level information (Rogers et al., 2020), and that using different token positions for classification results in only minor differences in accuracy (Goyal et al., 2020). Goyal et al. (2020) also found that embedding vectors in the final layer exhibit high cosine similarity due to information mixing through self-attention.

Motivated by these findings, we explore incremental modifications to the BERT base architecture for sequence classification, aiming to enhance its performance on downstream tasks. We specifically focus on improving the informativeness of the `[CLS]` token by (i) incorporating more *width* information of the whole sequence, and (ii) incorporating more *depth* information from additional layers. In the end we find that a mixture of these approaches leads to the best results.

**Contributions.**

1. We introduce MaxPoolBERT, an effective extension to BERT that enriches the `[CLS]` token representation using max-pooling and attention mechanisms across layers and tokens.

2. We systematically evaluate three architectural variants that incorporate *width* (token-level) and *depth* (layer-level) information into the `[CLS]` embedding.

3. We show that our proposed approach improves fine-tuning performance on 7 out of 9 GLUE tasks and achieves an average gain of 1.25 points over the BERT base baseline.

4. We demonstrate that MaxPoolBERT is particularly effective in low-resource scenarios, providing improved stability and accuracy where training data is limited.

All of our models will be made publicly available after the review process.

## 2   RELATED WORK

Much research has been done dedicated to improving and optimizing BERT's training process through architectural modifications and fine-tuning strategies. Below, we discuss advancements in fine-tuning stability, text representations, model enhancements, and training efficiency. Our work falls within the branch of research aimed at optimizing BERT's representation to enhance downstream classification results, with a particular focus on augmenting the informativeness of the `[CLS]` token.

**Stabilized BERT Fine-Tuning.**    The pre-training and fine-tuning paradigm for language models such as BERT (Devlin et al., 2019) has led to significant improvements across a wide range of NLP tasks while keeping computational costs manageable. However, fine-tuning remains unstable due to challenges like vanishing gradients (Mosbach et al., 2021) and limited dataset sizes (Zhang et al., 2021). Several studies have proposed techniques to address this instability.

Zhang et al. (2021) explore re-initializing BERT layers before fine-tuning, demonstrating that retaining all pre-trained weights is not always beneficial for fine-tuning. They also show that extending fine-tuning beyond three epochs improves performance. Hao et al. (2020) examine how fine-tuning affects BERT's attention, finding that higher layers change significantly while lower layers remain stable. They propose a noise regularization method to enhance stability. Mosbach et al. (2021) identify high learning rates as a key issue that cause fine-tuning instability. They propose using small learning rates with bias correction and increasing training iterations until nearly zero training loss is achieved. Hua et al. (2021) introduce Layer-wise Noise Stability Regularization which further stabilizes fine-tuning through regularization. Xu et al. (2023) propose self-ensemble and self-distillation mechanisms that enhance fine-tuning stability without requiring architectural changes or external data.

Our method, while not explicitly targeting stability, contributes to more robust performance especially on low-resource tasks by enabling the `[CLS]` token to integrate a broader context via pooling and attention. We analyze fine-tuning stability of our variants in Section 5.2.

**Faster and More Efficient Training.**    In addition to stabilization, architectural enhancements have been introduced to boost BERT's efficiency and effectiveness. Goyal et al. (2020) propose to eliminate tokens after fine-tuning, to reduce the time of inference. They discovered that the token representations in the highest layer of BERT base carry similar information.

Recently, Warner et al. (2024) introduce ModernBERT, an updated version of BERT with an increased sequence length of 8192. ModernBERT incorporates architectural improvements such as GeGLU activations (Shazeer, 2020), Flash Attention (Dao et al., 2022), and RoPE embeddings (Su et al., 2024).

While other approaches improve the input embedding size of BERT (Nussbaum et al., 2024) or refine the pre-training process for GPU's (Geiping & Goldstein, 2023; Portes et al., 2023; Izsak et al., 2021), our work specifically concentrates on optimizing the `[CLS]` token during fine-tuning, leveraging the information captured in BERT's layers after pre-training.

**Improved BERT Fine-Tuning and Representation Learning.** Lastly, several approaches refine BERT's classification capability through optimized fine-tuning strategies and enriched sentence representations - areas that align closely with our approach (see also Stankevičius & Lukoševičius (2024) who provide a comprehensive survey of methods for extracting sentence-level embeddings from BERT).

Toshniwal et al. (2020) systematically compared different text span representations using BERT, and found that max-pooling performs quite well across tasks, though its effectiveness varies. Bao et al. (2021) construct sentence representations for classification by selecting meaningful n-grams and combining sub-tokens of a pre-trained BERT model into span representations using a max-pooling approach. In contrast, our method does not require span selection or input modification, and applies pooling and attention directly to hidden states during fine-tuning.Hu et al. (2024) introduce a flexible BERT architecture with dynamic width and depth that adapts the number of attention heads, hidden size, and number of layers at inference time using knowledge distillation. Our approach does not alter the base architecture, instead we enrich the fixed-size `[CLS]` embedding to boost classification performance.

Chang et al. (2023) introduce Multi-CLS BERT, a framework that modifies pre-training and adds multiple `[CLS]` tokens to the sequence for fine-tuning. We achieve comparable results on the GLUE benchmark without altering the pre-training setup. Chen et al. (2023) present HybridBERT, which incorporates a hybrid pooling network and drop masking during pre-training to accelerate training and improve downstream accuracy. While HybridBERT combines multiple pooling strategies (mean, max, and attention) to replace the `[CLS]` token, we retain the original `[CLS]` embedding and instead enrich it through architectural refinements such as an additional multi-head attention layer and optional sequence-wide pooling. This allows our method to be applied to any pre-trained BERT-like model without re-training, with particular benefits observed on tasks with limited training data.

Recently, Galal et al. (2024) explore aggregation techniques such as mean pooling and self-attention on output embeddings for Arabic sentiment analysis. They show that freezing BERT during fine-tuning can boost performance. Our method can be combined with such techniques but focuses on improving the `[CLS]` pathway, especially under low-resource conditions.

Lastly, Lehečka et al. (2020) propose modifying BERT's output pooling strategy to improve large-scale multi-label text classification. Specifically, they replace the `[CLS]` token with combined mean and max pooling over the final hidden states of all tokens, arguing that this captures richer semantic information for classification. While their method entirely discards the `[CLS]` embedding, our approach retains it and enhances its contextual richness by integrating sequence-wide information via additional architectural layers during fine-tuning.

## 3 REFINING THE `[CLS]` TOKEN

It has been shown that other token representations in the layers of BERT also capture sentence-level representations (Rogers et al., 2020). We investigate whether the informativeness of the `[CLS]` token embedding can be further enhanced during fine-tuning, to improve downstream classification results. To do this, we include more *depth* information from other BERT layers and also more *width* information from other tokens within the sequence. We study different versions of fine-tuning BERT for sequence classification tasks. All variants are described below.

### 3.1 PRELIMINARIES

**Final-Layer `[CLS]` Representation.** As a baseline we use the `[CLS]` token of the final encoder layer of a fine-tuned vanilla BERT base model (Devlin et al., 2019) for classification (see Figure 2a). Recall that a single layer of BERT can be written as

$$f_i : \mathbb{R}^{T \times d} \to \mathbb{R}^{T \times d}, \tag{1}$$

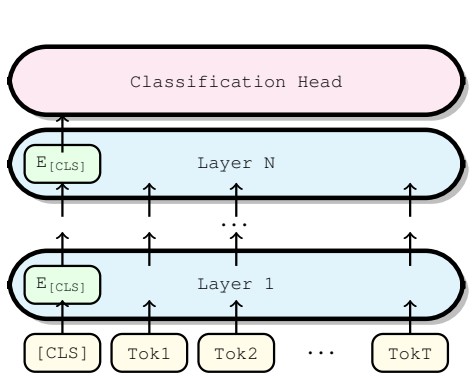

(a) **Baseline.** Plain vanilla BERT for sequence classification, where the embedding of the [CLS] token of the final layer is used as input for the classification head.

(b) **Max$_{\text{CLS}}$.** A max-pooling operation is applied on the [CLS] tokens of the last $k$ layers before classification.

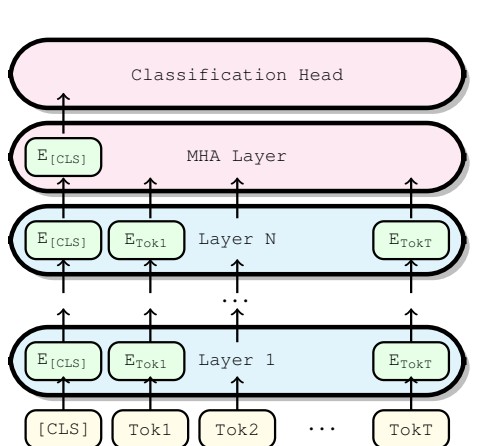

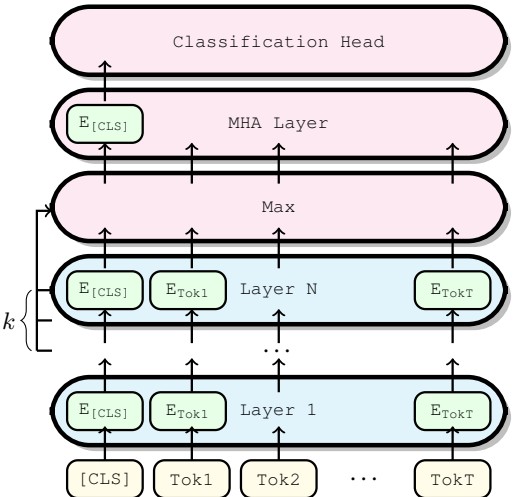

(c) **MHA.** An additional multi-head attention layer allows the [CLS] token to attend to all tokens of the last layer.

(d) **Max$_{\text{Seq}}$ + MHA.** A max-pooling operation on the whole sequence is combined with an additional MHA layer.

Figure 2: **Comparison of four BERT architectures for sequence classification.** *(Left above)* Classical BERT for sequence classification architecture. *(Right above)* Applying max-pooling on the token embeddings of the [CLS] token over the last $k$ layers. *(Left below)* Adding an additional MHA layer before classification. *(Right below)* MaxPoolBERT architecture: After the Nth layer (N = 12 for BERT base), we apply a sequence-wide max-pooling operation over the last $k$ layers (we used $k = 3$). The [CLS] token can then attend to every token after the max-pooling and the resulting [CLS] token embedding is used for classification.

where $i$ indicates the layer number (BERT base has 12 layers), $T$ is the number of tokens, and $d$ is the dimensionality of each token vector. We denote the values of the intermediate layers by $y^{(i)}$:

$$y^{(1)} = f_1(x), \qquad y^{(i+1)} = f_{i+1}(y^{(i)}). \tag{2}$$

The classification token of each layer is the first token, i.e., for a sequence of tokens $y^{(i)} = [t_{1i}, ..., t_{Ti}]$ in the $i$th layer,

$$[\text{CLS}]_i = t_{1i} \in \mathbb{R}^{1 \times d}. \tag{3}$$

The embedding of the [CLS] token serves as the input for the classification head $c$, which we have choosen to be a simple linear layer without an activation function, since we are just interested in the plain expressiveness of the refinement (instead of adding tanh as in the original BERT implementation):

$$c : \mathbb{R}^{1 \times d} \to \mathbb{R}. \tag{4}$$

Thus, the baseline model for sequence classification can be written as:

$$(c \circ \text{CLS} \circ f_{12} \circ \cdots \circ f_1) : \mathbb{R}^{T \times d} \to \mathbb{R} \tag{5}$$

for BERT base with 12 layers.

**Max-pooling operation.** The final layers of a BERT model are known to contain the task-specific information. In order to utilize not only the last layer but several layers, we have to define a flexible maximizing operation, that can work with several sequences of vectors. For this, we write $\Theta_t^{(k)} \in \mathbb{R}^{k \times t \times d}$ for the tensor that contains the first $t$ token vectors (each $d$ dimensional) of the last $k$ layers. For instance, $\Theta_1^{(1)} \in \mathbb{R}^{1 \times 1 \times d}$ is the [CLS] token, and $\Theta_1^{(k)} \in \mathbb{R}^{k \times 1 \times d}$ collects the token vectors the [CLS] token of the last $k$ layers. Similarly, $\Theta_T^{(1)} \in \mathbb{R}^{1 \times T \times d}$ contains all token vectors of the last layer, and $\Theta_T^{(k)} \in \mathbb{R}^{k \times T \times d}$ all token vectors of the last $k$ layers.

Next, we define an element-wise max-pooling operation that maximizes over the first dimension, i.e.,

$$\max : \mathbb{R}^{k \times t \times d} \to \mathbb{R}^{t \times d}. \tag{6}$$

Written as Pytorch[1] code, the operation is `torch.max(Theta, dim=1)` for $b$-sized minibatches of shape $b \times k \times t \times d$.

**Mean-pooling operation.** Several studies indicate, that max-pooling seems to be a stable choice to aggregate information into a single sentence representation. In the experimental section (Section 4), we also consider mean-pooling to challenge these results. For this, we apply an element-wise mean-pooling operation

$$\text{mean} : \mathbb{R}^{k \times t \times d} \to \mathbb{R}^{t \times d} \tag{7}$$

on every vector of our $k$ chosen layers (defined analogously as the max-pooling operation).

### 3.2 DEPTH-WISE [CLS] POOLING (MAX_CLS)

To use the *vertical* information (i.e., more depth) as one possible improvement for BERT's fine-tuning, we take information from the last $k$ layers (instead of only from the last layer): we extract the last $k$ [CLS] embeddings $[[\text{CLS}]_{12-k+1}, \ldots, [\text{CLS}]_{12}]$ which corresponds to $\Theta_1^{(k)} \in \mathbb{R}^{k \times 1 \times d}$ (using the notation of the previous paragraph). Then we apply the element-wise max-pooling operation on the extracted tokens (see Figure 2b).

### 3.3 TOKEN-WISE ATTENTION VIA ADDITIONAL MHA LAYER (MHA)

The orthogonal way to enrich the information in the [CLS] token, is to consider *horizontal* information (i.e., more width, see Figure 2c). For this, we include *all* tokens of the last layer. To obtain a single vector, we employ an additional multi-head attention (MHA) layer on the encoder output, but compute the attention only for the [CLS] token. We write the MHA as (see Vaswani et al., 2017),

$$\text{MHA(Q,K,V)} = [\text{head}_1, \ldots, \text{head}_h] W_0 \tag{8}$$

where the heads are defined as

$$\text{head}_s = \text{Attention}(Q W_s^Q, K W_s^K, V W_s^V). \tag{9}$$

Using the standard BERT base model with 12 layers, we have $Q = [\text{CLS}]_{12}$ and $K = V = y^{(12)}$. Through the attention mechanism, the [CLS] token can attend to all other tokens once more before classification. Note that the additional MHA layer is not part of the pre-training process and is added and initialized before the fine-tuning process. We use the default initialization of the Pytorch[1] multi-head attention implementation which is a Xavier uniform initialization (Glorot & Bengio, 2010). For the number of attention heads we choose $h = 4$.

---

[1]https://pytorch.org/

## 3.4 SEQUENCE-WIDE POOLING WITH MHA ($\text{MAX}_{\text{SEQ}}$ + MHA & $\text{MEAN}_{\text{SEQ}}$ + MHA )

Finally, we combine the additional *depth* and *width* information of $\textbf{Max}_{\textbf{CLS}}$ and **MHA** by extending the max-pooling operation to the whole sequences of the last $k$ layers by using $\max(\Theta_T^{(k)}) \in \mathbb{R}^{k \times T \times d}$. We call this setup $\textbf{Max}_{\textbf{Seq}}$**+ MHA**, since the maximum is now along the whole sequence and the additional MHA layer aggregates the pooled information. We call this approach **MaxPoolBERT** in the following. As a variant, we replaced max pooling with mean pooling. We report the results for mean pooling with an additional MHA layer as $\textbf{Mean}_{\textbf{Seq}}$**+ MHA**.

| Parameter | Value |
|---|---|
| learning rate | 2e-5 |
| epochs | 4 |
| batch size | 32 |
| warmup ratio | 0.1 |
| weight decay | 0.01 |

Table 1: Hyperparameters used for all fine-tuning experiments.

## 4 EXPERIMENTS

In order to evaluate each previously presented modification of the BERT architecture for sequence classification, we fine-tune each model on different classification tasks of the GLUE benchmark and compare the results. As a baseline, we use a standard BERT base model (Devlin et al., 2019). In addition, we assess the generalizability of our approach by applying it to a BERT variant, namely RoBERTa base (Liu et al., 2019).

### 4.1 DATASETS

The General Language Understading Evaluation (GLUE) benchmark (Wang et al., 2018) is a well known benchmark for natural language understanding (NLU) and natural language inference (NLI) tasks. We evaluate on the following 9 tasks:

- **CoLA** (Corpus of Linguistic Acceptability (Warstadt et al., 2019)): 10,657 sentences from linguistic publications, annotated for grammatical acceptability (*acceptable* or *unacceptable*).

- **MRPC** (Microsoft Research Paraphrase Corpus (Dolan & Brockett, 2005)): 5,800 sentence pairs from news source, annotated for paraphrase identification (*equivalent* or *not equivalent*).

- **QNLI** (Question NLI): an NLI dataset derived from the Stanford Question Answering Dataset (SQuAD) (Rajpurkar et al., 2016) containing question paragraph pairs. The task is to predict if the question is answered by the given paragraph (*entailment* or *no entailment*).

- **MNLI** (Multi-Genre NLI (Williams et al., 2018)): includes 433,000 sentence pairs, annotated with three different indicators for entailment (*neutral*, *contradiction* or *entailment*). MNLI includes both matched (in-domain) and mismatched (cross-domain) sections.

- **SST-2** (The Stanford Sentiment Treebank (Socher et al., 2013)): 215,154 phrases annotated for sentiment analysis (*positive* or *negative*).

- **STS-B** (Semantic Textual Similarity Benchmark (Cer et al., 2017)): 8,630 sentence pairs annotated with a textual similarity score (*from zero to five*).

- **RTE** (Recognizing Textual Entailment (Dagan et al., 2006)): 5,770 sentence pairs annotated for entailment recognition (*entailment* or *no entailment*).

- **QQP** (Quora Question Pairs): 795,000 pairs of questions from Quora, annotated for semantical similarity (*duplicate* or *no duplicate*).

- **WNLI** (Winograd NLI Levesque et al., 2012): 852 sentence pairs annotated for textual entailment (*entailment* or *no entailment*).

| Model | CoLA | MRPC | | QNLI | MNLI | | SST-2 | STS-B | RTE | QQP | | WNLI |
|---|---|---|---|---|---|---|---|---|---|---|---|---|
| | MCC | Acc. | F1 | Acc. | m | mm | Acc. | Sp. | Acc. | Acc. | F1 | Acc. |
| Train Size | 8.5k | 3.7k | 3.7k | 105k | 393k | 393k | 67k | 5.75k | 2.5k | 364k | 364k | 634 |
| BERT base | 53.59 | 82.43 | 87.49 | 90.96 | 84.27 | 84.57 | 92.55 | 88.47 | 63.42 | 90.65 | 87.40 | 49.77 |
| Max$_{CLS}$ | 55.32 | 83.66 | 88.5 | **91.15** | 84.22 | 84.55 | 92.62 | **88.97** | 63.06 | 90.59 | 87.33 | 50.23 |
| MHA | **55.88** | 85.38 | 89.51 | 90.49 | **84.37** | **84.67** | 92.32 | 88.04 | 64.98 | 90.67 | **87.45** | **55.4** |
| Max$_{Seq}$+MHA | 55.35 | **85.95** | **89.78** | 90.73 | 83.82 | 84.24 | **92.74** | 88.22 | 66.06 | 90.59 | 87.32 | **55.4** |
| Mean$_{Seq}$+MHA | 55.10 | 85.62 | 89.66 | 90.86 | 83.78 | 84.2 | 92.51 | 87.91 | **66.67** | **90.68** | 87.41 | 54.46 |
| Δ | 2.29 | 3.52 | 2.29 | 0.19 | 0.24 | 0.1 | 0.19 | 0.5 | 3.25 | 0.03 | 0.05 | 5.63 |

Table 2: **Our proposed variants improve the performance over BERT base on GLUE validation tasks (average of 3 seeds).** The size of the training data set is highlighted in gray. We report Matthews correlation coefficient (MCC) for CoLA, accuracies for matched (m) and mismatched results (mm) for MNLI, and Spearman correlation (Sp.) for STS-B. Below we report the improvement from the best performing variant over the baseline as Δ.

| Model | CoLA ↓ | MRPC ↓ | QNLI ↓ | MNLI ↓ | SST-2 ↓ | STSB ↓ | RTE ↓ | QQP ↓ | WNLI ↓ |
|---|---|---|---|---|---|---|---|---|---|
| BERT Base | 6.34e-02 | 2.42e-02 | **2.08e-03** | **1.97e-03** | **1.99e-03** | 3.2e-03 | **1.78e-02** | 10.8e-04 | 5.86e-02 |
| Max$_{CLS}$ | 4.55e-02 | **2.02e-02** | 3.89e-03 | 2.73e-03 | 3.81e-03 | 3.8e-03 | 1.86e-02 | 9.26e-04 | 4.61e-02 |
| MHA | 4.3e-02 | 2.1e-02 | 5.69e-03 | 2.43e-03 | 3.63e-03 | 4.99e-03 | 1.86e-02 | 8.09e-04 | 4.61e-02 |
| Max$_{Seq}$ + MHA | **4.22e-02** | 2.18e-02 | 5.11e-03 | 4.45e-03 | 3.64e-03 | 4.63e-03 | 1.96e-02 | 7.87e-04 | 4.31e-02 |
| Mean$_{Seq}$ + MHA | **4.22e-02** | 2.03e-02 | 4.49e-03 | 5.04e-03 | 4.46e-03 | 4.55e-03 | 2.12e-02 | **7.46e-04** | **3.97e-02** |

Table 3: **Standard deviations for three fine-tuning runs with different random seeds.**

## 4.2 EXPERIMENTAL DETAILS

All experiments were run on a single NVIDIA A100 GPU. We used the Huggingface transformers and dataset libraries[2] to implement and train all of our models. Each model was fine-tuned three times with three different random seeds for four epochs. We report the mean of all runs and use the validation sets of all GLUE tasks for evaluation. Experimenting with different values for $k$ (the number of the considered layers), we found that $k = 3$ works best (see Appendix A.1.1). All others hyperparameters are listed in Table 1.

## 5 RESULTS

We report the results for all model variants in each task and analyze fine-tuning stability by measuring the standard deviation between runs with different seeds.

### 5.1 PERFORMANCE ACROSS GLUE TASKS

The performance of each of our four variants on the GLUE benchmark tasks is presented in Table 2. For each task, at least one variant achieves higher performance than the BERT baseline, indicating that our proposed methods for enriching the [CLS] token representation are effective. However, the magnitude of improvement varies across tasks.

| Model | GLUE avg. |
|---|---|
| BERT Base | 79.63 |
| Max$_{CLS}$ | 80.02 |
| MHA | 80.76 |
| Max$_{Seq}$+MHA | **80.85** |
| Mean$_{Seq}$+MHA | 80.75 |
| Δ | 1.25 |

Table 4: **Average performance across all GLUE tasks.** MaxPoolBERT shows a consistent gain over BERT base.

The **Max$_{CLS}$** variant, which applies max-pooling over the [CLS] token representations from the last $k$ layers, results in marginal to no improvement for most tasks. Notably, this variant achieves the best performance among all variants on QNLI and STS-B, suggesting that layer-wise max-pooling can be beneficial for certain task types. Both tasks incorporate semantic matching between two texts, thus both require nuanced understanding of sentence meaning.

The **MHA** variant introduces an additional MHA layer, allowing the final-layer [CLS] token to attend to the full sequence before classification. This variant consistently improves upon the baseline BERT model, indicating that this

---

[2]https://huggingface.co/

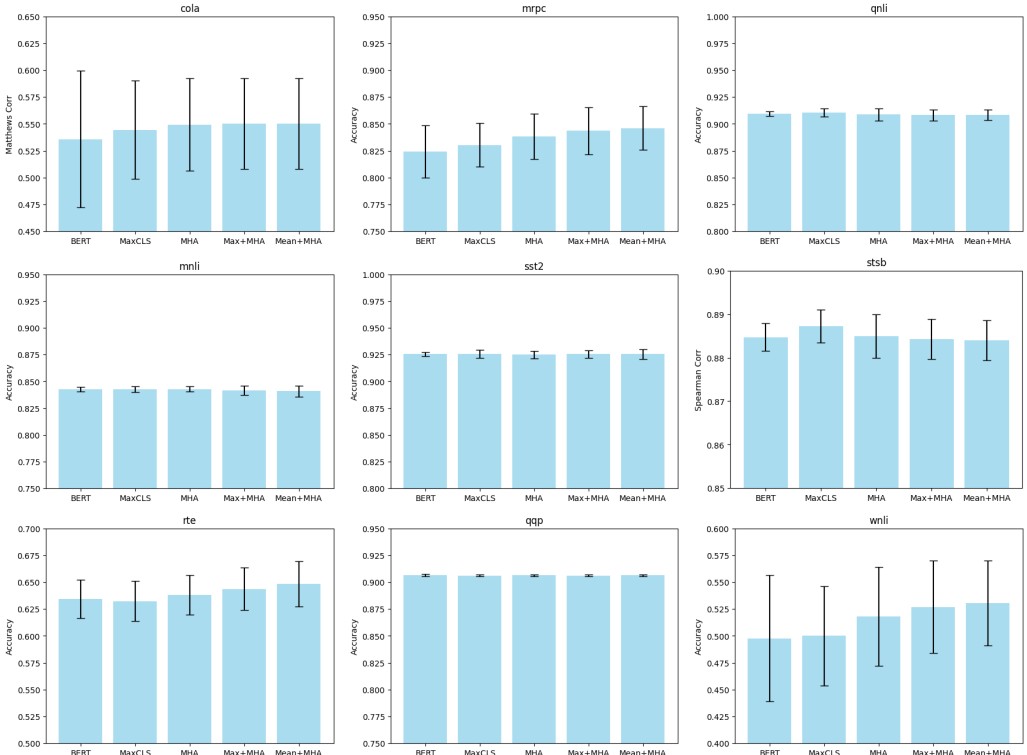

Figure 3: **Accuracies for the GLUE benchmark with error bars.** We show the standard deviation between three fine-tuning runs with three random seeds. Note that the y-axis is shifted but scaled equally across tasks.

extra attention step, effectively enhances the model's ability to integrate global context. The biggest improvement is observed on the WNLI dataset, which has the fewest training examples in the GLUE benchmark (634 training examples in total), suggesting that the added attention is particularly helpful in low-resource settings.

The **Max$_{Seq}$+MHA** variant combines token-wise max-pooling over the sequence with the additional MHA layer. This configuration shows the most consistent improvements, achieving a higher performance than the baseline in 7 out of 9 tasks. As shown in Figure 1, the largest improvements are again seen on datasets with limited training data, such as CoLA, MRPC, RTE and WNLI. These findings suggest that combining sequence-level pooling with attention further enhances robustness in low resource settings.

Overall, **Mean$_{Seq}$+MHA** performs similarly well as max-pooling, on RTE it even achieves higher performance than max-pooling. In the end, max-pooling seems to be a better choice as mean-pooling as it works better for most GLUE tasks.

| Model | GLUE avg. |
|---|---|
| RoBERTa Base | **82.62** |
| Max$_{Seq}$+MHA | 82.6 |
| Δ | -0.02 |

Table 5: **Average performance across all GLUE tasks.** MaxPoolBERT shows a gain over RoBERTa base.

For clarity, Table 4 shows the average performance of each model variant across all GLUE tasks. The **Max$_{Seq}$+ MHA** variant, which we call **MaxPoolBERT**, achieves the highest overall average. While the average improvement over the baseline is 1.25 points, individual tasks show greater improvements.

Additionally we apply the max-pooling and MHA combination to a BERT variant, namely RoBERTa (Liu et al., 2019). The results are depicted in Table 6. For RoBERTa, our max-pooling + MHA variant shows improvements on 4 out of 9 tasks but on average both models perform equally (see Table 5). RoBERTa is an optimized

version of BERT with a similar architecture, however the effects vary on this BERT variant. A significant improvement can only be observed on the CoLA dataset.

| Model | CoLA | MRPC | | QNLI | MNLI | | SST-2 | STS-B | RTE | QQP | | WNLI |
|---|---|---|---|---|---|---|---|---|---|---|---|---|
| | MCC | Acc. | F1 | Acc. | m | mm | Acc. | Sp. | Acc. | Acc. | F1 | Acc. |
| Train Size | 8.5k | 3.7k | 3.7k | 105k | 393k | 393k | 67k | 5.75k | 2.5k | 364k | 364k | 634 |
| RoBERTa Base | 54.96 | **88.56** | **91.64** | **92.29** | **87.15** | 86.97 | 93.2 | 89.78 | **73.29** | 90.84 | 87.64 | **56.34** |
| Max$_{Seq}$+MHA | **57.43** | 87.58 | 91.01 | 92.15 | 86.99 | **87.14** | **93.43** | **90.49** | 70.4 | **90.93** | **87.85** | 55.87 |
| Δ | 2.47 | -0.98 | -0.63 | -0.14 | -0.16 | 0.17 | 0.23 | 0.71 | -2.89 | 0.09 | 0.21 | -0.47 |

Table 6: **Performance on GLUE validation tasks for RoBERTa (average of 3 seeds).**

## 5.2 STABILITY ON LOW-RESOURCE TASKS

To assess fine-tuning stability, which is usually worse for smaller datasets (Devlin et al., 2019; Lee et al., 2020; Dodge et al., 2020), we run all experiments with three different seeds for each GLUE task. We report the mean accuracy across runs (for CoLA we report Matthews correlation coefficient, for STS-B we report Spearman rank correlation), and include error bars showing the standard deviation of these three runs (see Figure 3 and Table 3).

We observe that the stability in fine-tuning remains comparable across model variants for most datasets. However, improvements are observed for datasets with fewer training samples such as CoLA, MRPC, QQP and WNLI, where our variants exhibit reduced variability between runs. These findings suggest that our proposed modifications improve robustness in the low-sample regime.

## 6 CONCLUSION

We introduced MaxPoolBERT, a lightweight yet effective refinement of BERT's classification pipeline that improves the representational quality of the `[CLS]` token for the BERT base model. Our method leverages max-pooling across layers and tokens, and introduces a multi-head attention layer that allows the `[CLS]` token to re-aggregate contextual information before classification. These modifications require no changes to pre-training and add minimal overhead to fine-tuning.

Empirical results on the GLUE benchmark demonstrate that MaxPoolBERT outperforms standard BERT base across most tasks, with especially strong improvements in low-resource settings. This suggests that BERT's native use of the final-layer `[CLS]` embedding underutilizes available information and that small architectural additions can enhance generalization without sacrificing efficiency.

## LIMITATIONS

While MaxPoolBERT improves downstream performance, several limitations remain:

- **No task-specific tuning.** Our experiments use shared hyperparameters across tasks. Further gains could be possible with task-specific settings for pooling depth, attention heads, or training schedules.

- **Model size and generalization.** Our work focuses on BERT base. We also examined one BERT variant but were not able to demonstrate an advantage of applying max-pooling + MHA for RoBERTa.

- **Scope of evaluation.** We focus on sentence-level classification tasks in GLUE. The applicability of our approach to other tasks, such as token classification, generation, or cross-lingual transfer, is not yet evaluated.

In the future we aim to further investigate how to optimize the fine-tuning of small BERT models. While larger models often yield better performance, smaller models are crucial in real-time or resource-constrained environments. The BERT training paradigm following pre-training and fine-tuning has been predominant for several years and is widely used, so it is important to study whether further improvements can be made through small changes to this learning paradigm.

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

# A   APPENDIX

## A.1   ABLATIONS

We conduct ablation studies to test different modifications of our architectures. We describe all experiments and report their results in the following.

### A.1.1   CHOICE OF $k$

We experiment with the choice of $k$ for the max-pooling layer on the smaller GLUE datasets (CoLA, MRPC and RTE) and report the results in the following in Table 7. Because we average over three runs with different random seeds, the choice of k does not have an immense influence on performance, but it is apparent that $k = 3$ is the best choice on the data sets tested.

### A.1.2   TIME DIFFERENCE

To evaluate differences in fine-tuning and inference time, we measured the time to fine-tune both standard BERT and MaxPoolBERT for four epochs on the MRPC dataset on a single A100 GPU. We also measured the inference time on the MRPC validation set for both model variants. Fine-tuning BERT took 289.298 seconds (approx. 72.324 seconds per epoch), inference on the validation set took 2.9935 seconds. In contrast, fine-tuning MaxPoolBERT on the MRPC dataset takes: 294.504 seconds (approx. 73.626

|          | CoLA  | MRPC  |       | RTE   |
|----------|-------|-------|-------|-------|
|          | Acc.  | Acc.  | F1    | Acc.  |
| $k = 1$  | 54.85 | 83.58 | 88.44 | 63.18 |
| $k = 2$  | 55.76 | 85.21 | 89.29 | 65.34 |
| $k = 3$  | 55.35 | **85.95** | **89.78** | **66.06** |
| $k = 4$  | **56.42** | 85.29 | 89.27 | 65.34 |
| $k = 6$  | 55.65 | 85.13 | 89.25 | 65.7  |
| $k = 12$ | 55.41 | 85.21 | 89.17 | 65.46 |

Table 7: **Effect of max-pooling depth $k$ on small GLUE tasks.** $k = 3$ generally yields best results.

seconds per epoch). Inference time on the validation set is 3.0284 seconds. That is a difference of approximately 1.3 seconds per epoch of fine-tuning and 0.035 seconds difference for inference, which is neglectable.

