# OpenReview forum: "MaxPoolBERT: Enhancing BERT Classification via Layer- and Token-Wise Aggregation"
_ICLR.cc/2026/Conference — ICLR 2026 Conference Withdrawn Submission_

### Official Review · Reviewer_UDbs · 2025-10-26

**Soundness:** 2
**Presentation:** 3
**Contribution:** 1
**Rating:** 2
**Confidence:** 4

**Summary:**

This paper describes a simple extension to a BERT-based classification model, proposing three variants that replace the use of only the final-layer [CLS] token representation for classification tasks.
Three modifications are (i) max-pooling the [CLS] token across multiple layers, (ii) enabling the [CLS] token to attend over the entire final layer using an additional multi-head attention (MHA) layer, and (iii) combining max-pooling across the full sequence with MHA.
Experiments on the GLUE benchmark show that for all datasets, at least one of the proposed variants outperforms the vanilla version.

**Strengths:**

The paper is well-structured and easy to follow; the proposed modification is clearly described and easy to understand.

**Weaknesses:**

The empirical results are not strong enough to convince me that the proposed modification is useful. On 5 out of 9 datasets, the performance gap between the best-performing variant and vanilla BERT is less than 0.5. There is no clear winner among the proposed variants, making it unclear which one to use. Additionally, the method requires extra hyperparameter tuning (e.g., selecting the last $k$ layers).

**Questions:**

Suggest adding a column to Table 2 that shows the average results for each model, rather than reporting only the improvement of the best-performing variant over the baseline.

---

### Official Review · Reviewer_gwPt · 2025-10-28

**Soundness:** 2
**Presentation:** 3
**Contribution:** 4
**Rating:** 4
**Confidence:** 3

**Summary:**

This paper tries to address a problem in BERT that it only uses the [CLS] token for classification, while a lot of information in other tokens and other layers is wasted.
The authors propose some small changes to the BERT architecture to fix this. Their methods include the following two parts. Max_CLS means doing a max-pooling over the [CLS] token from the last few layers, and MHA is adding a new multi-head attention layer at the end, so the [CLS] token can collect information from other tokens again. Their final model calls MaxPoolBERT, which is a combination of the two methods above. It first does max-pooling over all tokens in the last few layers, and then uses the new MHA layer.
They test on the GLUE benchmark.  The experiments on GLUE benchmark show their MaxPoolBERT method works good for the bert-base-uncased model, especially when the training data is not much (low-resource datasets like MRPC, RTE). However, an important result is that their method does not work for RoBERTa-base. It actually makes RoBERTa a little bit worse.

**Strengths:**

1.Good Clarity and Motivation. The paper is easy to read and the motivation is very clear. Why we need to improve the [CLS] token representation is well explained with support from other papers.

2.Focus on Low-Resource and Stability. For the BERT-base model, the results are strong. On some small datasets like MRPC and RTE, the improvement is quite big, which shows the method has some real effect in low-resource situations. They also show that their model has a smaller standard deviation across different runs, which means it is more stable.

3.Systematic Experiments. The authors test their ideas step-by-step. They show results for Max_CLS only, MHA only, and then the combined model. This is a good way to show where the performance gain comes from.

**Weaknesses:**

1.Failure to Generalize. This is the most serious weakness. The method improves BERT-base but slightly hurts RoBERTa-base. This strongly suggests that it is not a general method for improving similar models. It is more like a "patch" that fixes a specific weakness in the original BERT-base model. The reason why it fails on RoBERTa should be investigated.

2.Limited Novelty. Max-pooling and attention are standard tools. The paper's contribution is more of an incremental engineering improvement by combining existing components.

3.Lack of Comparison. The experiments are only on GLUE classification tasks and mostly focus on BERT-base. It would be stronger if they showed it works on other models or other types of tasks. Also, the paper only compares with the default fine-tuning of bert-base-cased, which is not enough. They should have included more baselines.

**Questions:**

Why do you think your method failed on RoBERTa? Could you please do more analysis to find the reason?

Could you please strengthen your experiments by adding more baselines?

Did you try your method on other models? It would be interesting to see if it helps other small models that are also widely used.

---

### Official Review · Reviewer_juDy · 2025-10-28

**Soundness:** 2
**Presentation:** 2
**Contribution:** 2
**Rating:** 4
**Confidence:** 3

**Summary:**

This paper proposes MaxPoolBERT, a lightweight extension of BERT that enhances the [CLS] token’s representational quality for classification by aggregating layer-wise and token-wise information via max-pooling and multi-head attention layer. It requires no new pre-training, adds minimal overhead, and outperforms the standard BERT model across 7/9 GLUE tasks.

**Strengths:**

- The overall idea is simple and straightforward.
- The paper is easy to follow. The proposed method is explained in detail and is easy to reproduce.

**Weaknesses:**

- The results are only evaluated on BERT and RoBERTa, with testing conducted exclusively on the GLUE benchmark.
- The idea of enhancing the [CLS] token has been extensively studied in the field of Vision Transformers (ViTs) and their variants. However, due to the variable input lengths of NLP tasks, different strategies may exhibit performance variations across distinct tasks. It would be valuable to further evaluate the proposed method on additional benchmarks.
- I believe that conducting experiments solely on BERT and the GLUE benchmark is far from sufficient, given the current state of development in the field. While I acknowledge that exploring ways to enhance the [CLS] token can yield certain benefits, this line of research is relatively outdated. Moreover, the work primarily constitutes an engineering attempt. There is no intuitive evidence of theoretical foundations, insightful theoretical analysis, or the potential for generalization to more universal models in the proposed method.

**Questions:**

- Will the added multi-head attention layer slow down the inference speed compared to vanilla BERT?

---

### Official Review · Reviewer_FkzZ · 2025-11-02

**Soundness:** 1
**Presentation:** 2
**Contribution:** 1
**Rating:** 2
**Confidence:** 4

**Summary:**

This paper proposes a new pooling method for the BERT model. This method performs max pooling across layers and various positions in the sequence, and introduces an additional Multi-head attention layer to aggregate information in the sequence. The proposed method improves over the original BERT on the GLUE benchmark.

**Strengths:**

1. The paper is straightforward, though it could benefit from more in-depth analysis.

**Weaknesses:**

1. Several small evaluation sets, like RTE, show high variance that is higher than the performance gain reported in this paper.

2. This paper does not compare to other papers' improvements over BERT, but only compares to the Valina BERT.

3. The results of the BERT baseline  reported in this paper are lower than those of others.

4.  In the final layer, Cls token already attend to all tokens, and passes infromation from past layers through the  residual connection, why do we need addtional settings? This paper lacks suffient justifications, but only relies on huristics.

5. This paper only report performance on the glue benchmark.

**Questions:**

1. Why do you claim that the proposed method improves over Roberta base in the caption of Table 5 even the average scores of your method is lower than the roberta baseline?

---

### Note · Authors · 2025-11-13

**Comment:**

We would like to thank all the reviewers for taking the time to provide valuable feedback on our work. Although we recognize the potential of our approach, particularly in low-resource contexts, we acknowledge the criticism and will conduct further experiments before resubmitting the paper.

**Withdrawal Confirmation:**

I have read and agree with the venue's withdrawal policy on behalf of myself and my co-authors.